# Quantifying benefits of renewable investments for German residential Prosumers in times of volatile energy markets

**Jonas van Ouwerkerk** [1,2,3,4] ✉, **Mauricio Celi Cortés** [1,2,3,4], **Najet Nsir** [1,2,3,4], **Jingyu Gong** [1,2,3,4], **Jan Figgener** [1,2,3,4], **Sebastian Zurmühlen** [1,2,3,4], **Christian Bußar** [1,2,3,4] & **Dirk Uwe Sauer** [1,2,3,4,5]

The COVID-19 pandemic and the Russian invasion of Ukraine have led to unseen disruptions in the global energy markets since the end of 2021. Residential renewable investments like photovoltaic systems, battery home storage systems, and heat pumps are therefore gaining traction. However, the benefits of those technologies during the energy crisis and beyond have not been fully quantified yet. Therefore, in this study, we benchmark renewable investments for a broad variety of single-family homes by evaluating potential cost savings and emission reductions. In addition, the study considers the influence of recent political incentives and subsidies. The results show that photovoltaic systems are a no-regret investment decision, both economically and environmentally. At the climax of the energy crisis, a typical German household with a heat pump could save 1850 € and reduce equivalent $CO_2$ emissions by 250 g/kWh annually. Politically introduced price breaks on electricity and natural gas do not reverse this advantage. Furthermore, when owning an electric vehicle renewable investments are often more beneficial.

In the aftermath of the disruptions caused by the COVID-19 pandemic, there has been a substantial increase in energy prices in the European Union (EU) (Fig. 1). At the beginning of 2022, gas prices at the stock exchange were over 600% higher compared to January 2020[1]. This made power generation by gas-fired power plants significantly more expensive and caused electricity prices to rise by almost 500% in the same period of time[2,3]. The main drivers for this development were rising natural gas (NG) demand after the COVID-19 restrictions and geopolitical tensions between Russia and the EU[4,5]. These uncertainties on the NG markets led to lower filling of seasonal storage as usual before the winter[6,7]. The energy crisis was aggravated in early 2022 by the Russian invasion of Ukraine. Economic sanctions of Western countries against the Russian Federation and the subsequent reduced gas transfer caused further turbulence in the energy markets, with record-high Dutch TTF gas futures of over 700 €/MWh in mid-2022[1,8,9]. In addition to high average market prices, an increasing volatility in the energy markets has been observed ever since. The average daily price spread at the intraday electricity market grew from 62.7 €/MWh in 2019 to 322 €/MWh in 2022[10,11].

In order to increase resilience against rising energy prices, consumers can take individual actions. One main measure is to reduce the consumption of NG for space heating by lowering room temperatures.

[1]Center for Aging, Reliability and Lifetime Prediction of Electrochemical and Power Electronic Systems (CARL), RWTH Aachen University, Aachen, Germany. [2]Institute for Power Electronics and Electrical Drives (ISEA), RWTH Aachen University, Aachen, Germany. [3]Institute for Power Generation and Storage Systems (PGS), E.ON ERC, RWTH Aachen University, Aachen, Germany. [4]Jülich Aachen Research Alliance, JARA-Energy, Aachen, Germany. [5]Helmholtz Institute Münster (HI MS), IMD-4, Forschungszentrum Jülich, Jülich, Germany. ✉e-mail: batteries@isea.rwth-aachen.de

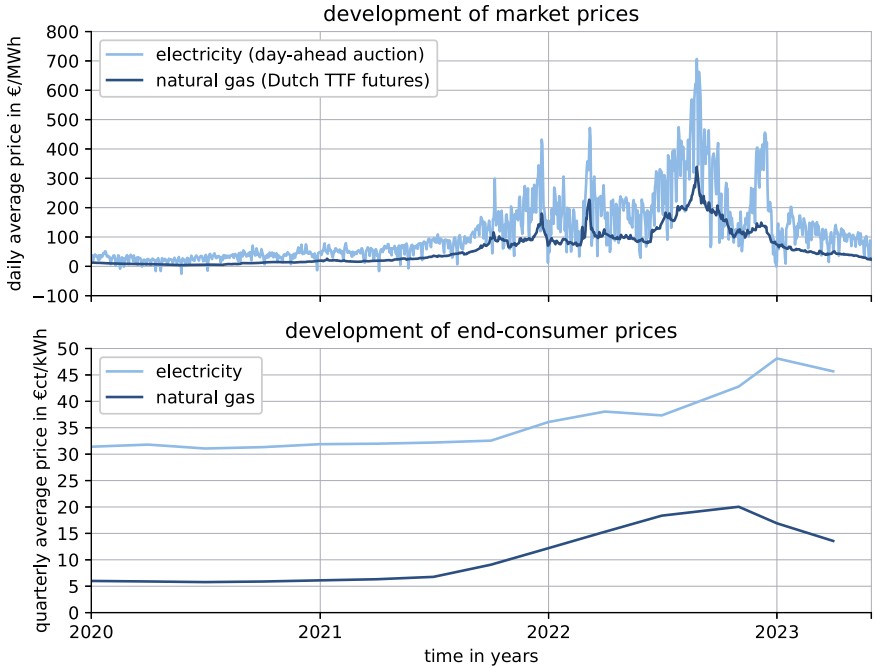

**Fig. 1 | Development of energy prices.** NG and electricity prices at the stock exchange and for German end-consumers from 2020 to mid-2023, based on data from refs. 1,2,12. For electricity day-ahead auction prices the average daily values are illustrated.

In 2022, households in Germany consumed about 12% less NG than usual considering normal year temperature corrections[7]. While at the beginning of 2022, these savings most likely were rather a first political or ethical response to the Russian invasion of Ukraine, with rising energy prices throughout 2022, the main objective of households became to save energy costs[6,7]. In addition to NG savings, electricity consumption showed a decline of 2.3% in 2022, compared to 2021[12]. Apart from energy savings, investments in renewable energies have become increasingly popular in Germany before and during the crisis. The number of residential solar photovoltaic (PV) systems in Germany has been constantly growing[13,14]. For home storage system (HSS), Figgener et al. estimate a market growth of 52% in 2022 in terms of deployed energy capacity[15]. With regard to heating, the market for heat pump (HP) systems in Europe shows an increasing trend with record high installations of over 2 Mio. in 2021[16].

While many studies investigate renewable investments for German residential Prosumers, they are mostly considering few household topologies with specific scope[17,18] or focusing rather on consumption behavior[19]. A study by Meyer et al. comparing conventional and renewable heating systems comes to the conclusion that most renewable low-emission solutions like the air-source HP are at least equally expensive or even financially beneficial[20]. According to Acke et al., households in Germany have achieved yearly energy cost savings of 1263 € with a PV system and 3614 € by combining PV and HP in 2022[17]. A holistic comparative analysis of the development of Prosumer energy costs in Germany, including the effects of the energy crisis, does not exist to the best of our knowledge.

Rising energy prices have far-reaching consequences for end-consumers in Germany, especially for low-income households[21–23]. About 600,000 households are now at risk of falling below the poverty threshold[23]. In order to mitigate the consequences of the crisis, the German Federal Government has decided on a number of measures to relieve the burden on consumers. For 2022, this includes a reduction in fuel tax, a public transport ticket for nine euros per month, one-time payments for people on low incomes, heating cost subsidies, and further tax relief[24]. For 2023, energy price breaks have been announced with fixed prices of 40 €ct/kWh for electricity and 12 €ct/

kWh for NG for 80% of the yearly demand taking the previous year as a reference[25]. Despite those far-reaching measures, it remains controversial if this is sufficient to effectively lower the impacts of the crisis for those who are most affected[26].

In this work, we quantify and compare renewable investments that can help to reduce energy costs for single-family homes (SFH) and, at the same time, contribute to climate mitigation. Furthermore, we analyze the influence of political measures to combat rising energy prices, including price breaks for electricity and NG, as well as considering value-added tax (VAT) exemptions for renewable investments. With our approach, we prove that renewable technologies usually offer a substantial cost savings and emission reduction potential for SFH, under consideration of the current political framework conditions.

## Results and discussion
### Savings potential with renewable investments in SFH
In our analysis we investigate possible savings that can be achieved with renewable investments in SFH. The savings potential is calculated by comparing the overall yearly system costs from the optimization of a variety of renewable SFH (H1–H4) with the results of the fossil standard household (H0-GAS). For yearly overall system costs, we consider annualized investments, fixed operational costs for maintenance, and yearly variable operating costs (see "Optimization approach" in "Methods"). The savings results for the different household topologies (see "Scenario set-up" in "Methods") are shown in Fig. 2. For each topology, we evaluate selected households with the aim of covering a wide range of typical constellations. A three-person household in a building stock constructed between 1979 and 1990 functions as a reference throughout the results analysis. For the analysis of savings, additional variations of this reference are considered. On the one hand, the number of household residents varies between one and six (Fig. 2a). On the other hand, we consider different construction years of the building for the reference three-person household (Fig. 2b).

**PV as no-regret option.** The savings results highlight that investing in solar PV is a no-regret investment decision for German households (H1-

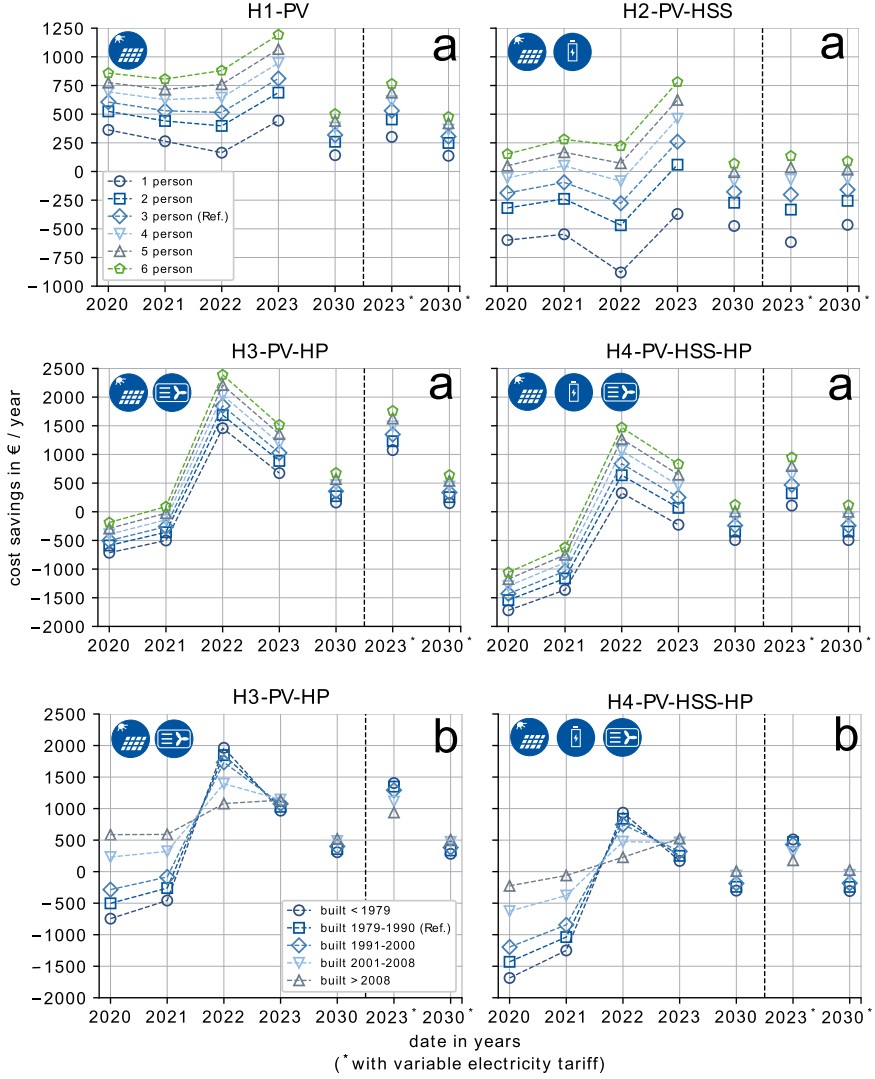

**Fig. 2 | Prosumer cost savings.** Savings that can be achieved for different SFH topologies, compared to a standard fossil household utilizing NG. **a** Variation of number of residents, for a building constructed between 1979 and 1990. **b** Variation of construction year of a building, for a three-person household.

PV in Fig. 2a). In all of the considered annual scenarios, a SFH with a solar PV installation is economically more attractive compared to the fossil standard household (H0-GAS) without such a system. For the reference three-person household, the savings are between 318 and 811 € (8–17%, Supplementary Fig. 7), depending on the considered year. From 2020 to 2021, decreasing feed-in tariffs reduce the savings of investing in solar PV. In 2022, despite high electricity prices due to the energy crisis, the savings further decrease for households with less than four residents as investment costs for solar PV increase. The decreasing trend reverses for all households in 2023 although investment costs remain on a high level. The increase can be explained with higher electricity prices and the VAT tax exemption for PV from 2023 on that is put into place by the German government. In addition, the elimination of the law that limits PV grid feed-in to 70% of the nominal power capacity increases feed-in profits. In comparison with the reference three-person SFH, the number of residents has a substantial influence on the savings. While a two-person household has 59–123 € less annual savings than a 3-person household, a four-person household increases the savings by 65–135 €. Furthermore, the savings of a one-person household are substantially lower as electricity consumption is low, and thus, the household can only utilize 12% of the total PV generation (Supplementary Fig. 3). The future outlook on the savings of residential PV for 2030 indicates a substantial decrease. Therefore, a

reevaluation of remuneration mechanisms should be carefully considered in the coming years when aiming for steady growth rates of residential solar PV.

**Battery home storage systems to increase autarky.** The savings with a HSS are, other than for solar PV systems, less intuitive (H2-PV-HSS in Fig. 2a). As a HSS is usually designed to increase the self-consumption of solar PV generation it only generates a benefit in combination with PV. In comparison with a SFH with solar PV only (H1-PV), the results indicate that a household with an additional HSS has 0.8 to 2.7 times lower cost savings in 2020. In addition, when comparing this topology to the fossil standard household the reference three-person household has lower savings between 95 and 187 € (2–5%) in 2020 and 2021. In 2022, this disadvantage further increases to 277 € as turbulence in the markets due to the energy crisis leads to a substantial increase in investment costs of HSS.

The picture changes in 2023 with further increasing energy prices and measures by the German federal government. The gap between a PV only household and one with HSS gets smaller as the battery storage reduces expensive grid supply. At the same time, investment costs are lower because of the VAT exemption. For the six-person household, the savings in 2023 with HSS (H2-PV-HSS) are still 412 € lower compared to the PV-only household (H1-PV). However, the

household now has potential annual savings of 781 € (10%) compared to the fossil standard household (H0-GAS). This highlights that a HSS effectively increases independence from the public grid and thus leads to higher robustness against rising energy prices during a crisis. When considering the savings results for a two-person and four-person SFH the increase in autarky highly depends on the energy consumption of the household (Supplementary Fig. 4). The higher the consumption (and the number of residents), the higher the effectiveness of the 9.4 kWh HSS. Overall, the savings of a 1–3 person SFH with HSS is substantially lower than for the fossil standard household. Therefore, for a SFH with fewer residents (1–3 person) the standard 9.4 kWh sized HSS can be considered too large. As a future outlook, the projection for 2030 indicates that a 9.4 kWh HSS in combination with solar PV is expected to be always less economically attractive than the fossil standard household (H0-GAS). Although investment costs for HSS can be expected to substantially decrease until 2030, electricity prices are projected to decrease as well. The current typical use case of HSS will thus become less attractive in the future, and other additional operating strategies, e.g., providing ancillary services, need to be exploited.

**Air-source heat pumps to combat gas price increase.** The strongest impact of the energy crisis on SFH energy consumption can be observed in the heating sector. The steep increase in the price of NG, thus a low ratio of electricity to NG price (see Table 1), turns the investment into a renewable heating system like a HP into a highly attractive investment (H3-PV-HP in Fig. 2a). Considering the reference three-person household in a building stock between 1979 and 1990, an air-source HP in combination with solar PV saves 1848 € (28%) in 2022, compared to a SFH utilizing NG for heating (H0-GAS). The high efficiency of the HP, in combination with a solar PV system, substantially increases the robustness of a SFH against rising energy prices. However, the savings results in 2020 indicate that before the energy crisis, a SFH constructed between 1979 and 1990 with an air-source HP usually had a financial disadvantage over a gas boiler (GB) system with higher annual overall system costs of 187–715 € (4–28%). In 2021, despite decreasing overall system costs, an air-source HP system only generates savings for households with more than five residents. The main reason is the high ratio of electricity to NG prices. However, with the energy crisis (2022 and 2023) this entirely changes. All households can now achieve substantial savings in comparison with the fossil standard household. In 2023, the savings are lower, which is caused by the price breaks, especially on NG by the German government. Furthermore, focusing on the construction year of the building (Fig. 2b) reveals that investing in energy efficiency is equally important than investing in renewable heating components. An energy-efficient building constructed after 2008 is substantially less affected by the gas price increase during the energy crisis (Fig. 2b). In addition, before the energy crisis (2020), these households can already achieve high annual savings of 586 € (27%) as they consume less energy and require smaller air-source HP systems which as a result reduces investment costs. Toward 2030, investment costs for air-source heat pump systems are projected to further decrease which further increases their pre-crisis level of savings (2020) in comparison with a GB.

**Combination of all renewable technologies financially less advantageous.** The savings results indicate that a combination of solar PV, HSS, and HP (H4-PV-HSS-HP in Fig. 2a) is economically less attractive than the HP-only (H3-PV-HP) topology. In 2020 and 2021, none of the SFH configurations (Fig. 2b) generates savings in comparison with the standard fossil fuel SFH (H0-GAS). This is mainly caused by the high investment costs of this topology. However, the picture again changes during the energy crisis (2022), with savings of 839€ (28%) for the standard three-person household (built 1979–1990). The HSS effectively increases the self-consumption of solar PV generation, which substantially lowers the consumption from

the grid (Supplementary Fig. 3). In addition, the projection for 2030 reveals that the fully equipped renewable SFH (H4-PV-HSS-HP) with less than 6 residents always is expected to have equal or even higher costs in comparison with the fossil standard household (H0-GAS). However, with the assumption that energy prices will not immediately fall back to the pre-crisis level, combining the three renewable technologies can be a feasible solution to decrease energy dependence for selected households.

**Influence of variable electricity tariffs on savings with renewable investments.** Under the assumption of a fixed electricity demand, flexible electricity tariffs are applied to selected years 2023 and 2030. From the results (see Fig. 2), we see that this can have a noticeable impact on possible savings. However, the picture varies between the different renewable topologies. For the topology with PV system only (H1-PV) and PV system plus HSS (H2-PV-HSS), the 2023 savings substantially decrease. This implies that switching to a flexible electricity tariff alone can already be beneficial for a standard fossil fuel SFH (H0-GAS). While the PV system can only substitute cheap electricity during noon, the HSS cannot gain a clear benefit by shifting PV generation to times in the evening with higher prices. The picture changes when including a HP acting as a flexible electricity demand (H3-PV-HP and H4-PV-HSS-HP). The flexibility of the HP allows to consume electricity from the grid at lower prices which increases savings in 2023 between 335 and 401 € for a building constructed between 1979 and 1990. However, for buildings with higher energy efficiency (built after 2001), this effect cannot be observed as the sizes of the HPs are smaller and thus less PV generation can be utilized (Supplementary Fig. 4). For future years (2030), the results show almost no differences when comparing fixed and flexible electricity tariffs. This is explained by the assumption of lower average electricity prices in 2030. Therefore, for flexible tariffs to be beneficial in future, consumers have to change their consumption behavior by shifting demands to times with lower prices.

**Influence of price breaks on savings with renewable investments.** The implemented price breaks on electricity and NG by the German government for 2023 aim at lowering the costs for households during the energy crisis. This potentially affects renewable investments, as especially SFH, with high electricity and NG consumption, could benefit from the regulation. The savings results in Fig. 3 confirm that for households with air-source HP the price breaks have a substantial impact on the savings that can be achieved with a renewable solution.

Nevertheless, for the typical three-person household investing in solar PV and an air-source HP economically remains highly beneficial with savings of 1028 € in 2023. However, this corresponds to a savings reduction of 40% in comparison with a scenario without price breaks. For investments into solar PV and or HSS, cost savings remain at 811 € and 260 €, respectively. The cost savings result for the PV-only household is less affected by the price break as the financial benefit mainly comes from the self-consumed PV generation. On the contrary, the cost savings for topologies with HSS are more affected as the savings of a HSS depends on the reduced grid consumption and costs. Considering that the price breaks only apply in 2023, the positive outlook for renewable investments in SFH will not be substantially affected. At the same time, despite price breaks, energy costs for households are effectively lowered by renewable investments in 2023.

**Emission reductions of renewable investments in SFH**
In this section, the equivalent $CO_2$ emission reduction potential for the different household topologies (see "Scenario set-up" in "Methods") is evaluated (Fig. 4a). The three-person household in a building stock constructed between 1979 and 1990 remains the reference throughout the analysis. Similarly to the cost savings results, the variations include the number of residents (Fig. 4a) and different construction years of

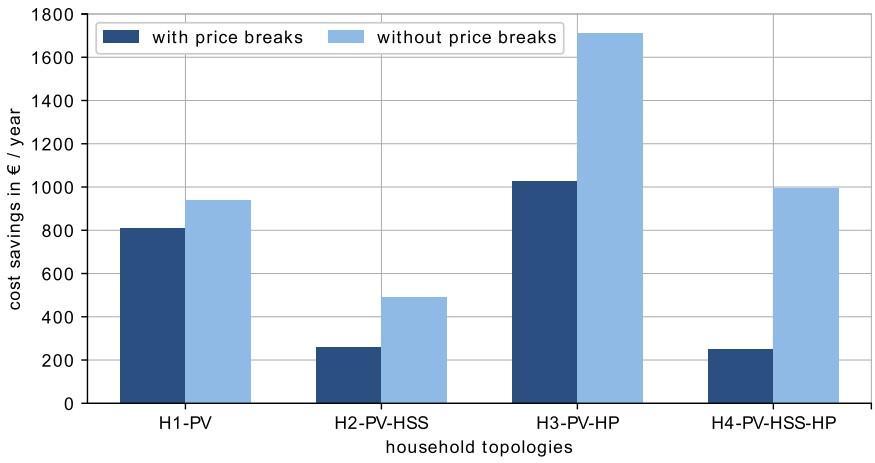

**Fig. 3 | Influence of price breaks.** Savings that can be achieved in 2023 for a typical three-person household (1979–1990) with or without price breaks on electricity and gas as proposed by the German government, for topologies H1–H4, compared to a standard fossil household.

the building stock (Fig. 4b). The equivalent $CO_2$ emissions include emissions from the public electricity grid, emissions for the provision of NG, and life cycle emissions of all relevant components including solar PV, HSS, HP, and inverters. For feed-in of electricity into the public grid, the saved emissions are positively credited/counted. All emission savings are normalized to the total energy consumption of the SFH including electricity, space heating (SH), and hot water (HW).

One primary observation is that the typical three-person household achieves reductions of equivalent $CO_2$ emissions per kWh of consumed energy across all renewable topology configurations (H1–H4) (Fig. 4a), compared to the standard fossil household. The higher the number of residents, the lower the emission reduction potential per kWh of consumed energy which increases with household size. For solar PV-only households, the emission savings are between 79 and 91 g/kWh of consumed energy (32–38%, Supplementary Fig. 8) per year (2020–2023) for the typical three-person household. For the same SFH, the component with the by far largest reduction potential is the air-source HP with a 236–251 g/kWh (73–89%) annual reduction. In comparison with a PV-only household, an air-source HP (H3-PV-HP) adds additional equivalent $CO_2$ emission reductions of at least 149 g/kWh per year (2020–2023). This is achieved by substituting fossil NG. Furthermore, the reduction potential is lower when a HSS is installed. In comparison with a solar PV-only household, the additional installation of a 9.4 kWh HSS (H2-PV-HSS) lowers the equivalent $CO_2$ reduction potential to only 32–36 g/kWh (2021–2023) (22–26%), for a typical three-person household. For HP-based systems, an additional HSS (H4-PV-HSS-PV) lowers reductions by 18–20 g/kWh (2020–2023) as a higher utilization of the HSS is achieved. The overall negative impact on reductions for a HSS is mainly caused by the production footprint, efficiency losses, and the temporal characteristics of emissions from the public grid.

The trend from 2020 to 2023 for the solar PV-only topology (H1-PV) reveals that with the energy crisis, the emission reduction potential increases due to higher emission factors for electricity from the public grid. On the heating side, identifying trends is less obvious with more than one influential factor. While emissions for the provision of NG decrease until 2023 as of the suspension of the supply of higher emitting NG from Russia during 2022[27–29] the emissions for electricity supply increase. For 2030, the reduction potential for the renewable topologies (H1–H4) toward the fossil SFH (H0-GAS) is projected to decrease. Although it is assumed that emissions for manufacturing of PV panels will substantially decrease toward 2030, with the projected ongoing decarbonization of the electricity grid households can gradually substitute less generation from fossil power plants. However, for all households, the local generation and use of green solar PV

electricity after all remains beneficial, especially with the electrification of the heating sector with HP systems.

For the estimation of equivalent $CO_2$ emission reduction potentials of SFH, the year of construction of the building is one of the most influential factors (Fig. 4b). The results for the typical three-person household indicate that the equivalent $CO_2$ emission savings potential per consumed kWh of energy per year with an air-source HP (H3-PV-HP) is 220–243 g/kWh higher for energy-efficient buildings (>2008) than for low efficient buildings (<1979) (2020–2023). The main reason for this is that the SH consumption of energy-efficient buildings is substantially lower; thus, a higher share of this consumption can be directly substituted by green electricity from solar PV. Nevertheless, even for energy inefficient buildings (<1979) the air-source HP achieves substantial savings of 200–210 g/kWh without and 196–205 g/kWh with HSS (2020–2023). In addition, the equivalent $CO_2$ emissions of more energy-efficient buildings are less dependent on the emission factor for the provision of NG (similar values from 2020–2023) and are more dependent on the emission factor of the electricity grid (lower value in 2030). The results for 2030 highlight that the gap in the reduction potential between high and low energy efficient buildings is projected to close with decreasing emissions of electricity from the public grid.

The introduction of flexible electricity tariffs has only a minor influence on the equivalent $CO_2$ emission reduction potential. A small reduction of 1–10 g/kWh can be observed for the topology with PV and HP (H3-PV-HP). This reduction comes from the flexible use of the HP. It is especially operating at times with low prices which also corresponds to times with high renewable generation and thus lower equivalent emissions.

## Influence of relevant modeling parameters

For validation and further interpretation of the results we evaluate the influence of different parameters (see Fig. 5). This includes the size of the solar PV system ("Influence of the PV size on the results" in "Results and discussion"), the consideration of SFH that own an electric vehicle (EV) ("Influence of owning an EV on the results" in "Results and discussion"), and potential investment costs increases as of the COVID-19 pandemic and the energy crisis ("Influence of component costs on the results" in "Results and discussion").

**Influence of the PV size on the results.** From the previous analysis, it becomes clear that the solar PV system has a substantial impact on the level of monetary savings that can be achieved. Therefore, to quantify the impact of the PV system on the results its size is increased from 8.7 kWp to 13.7 kWp, which corresponds to the average maximum

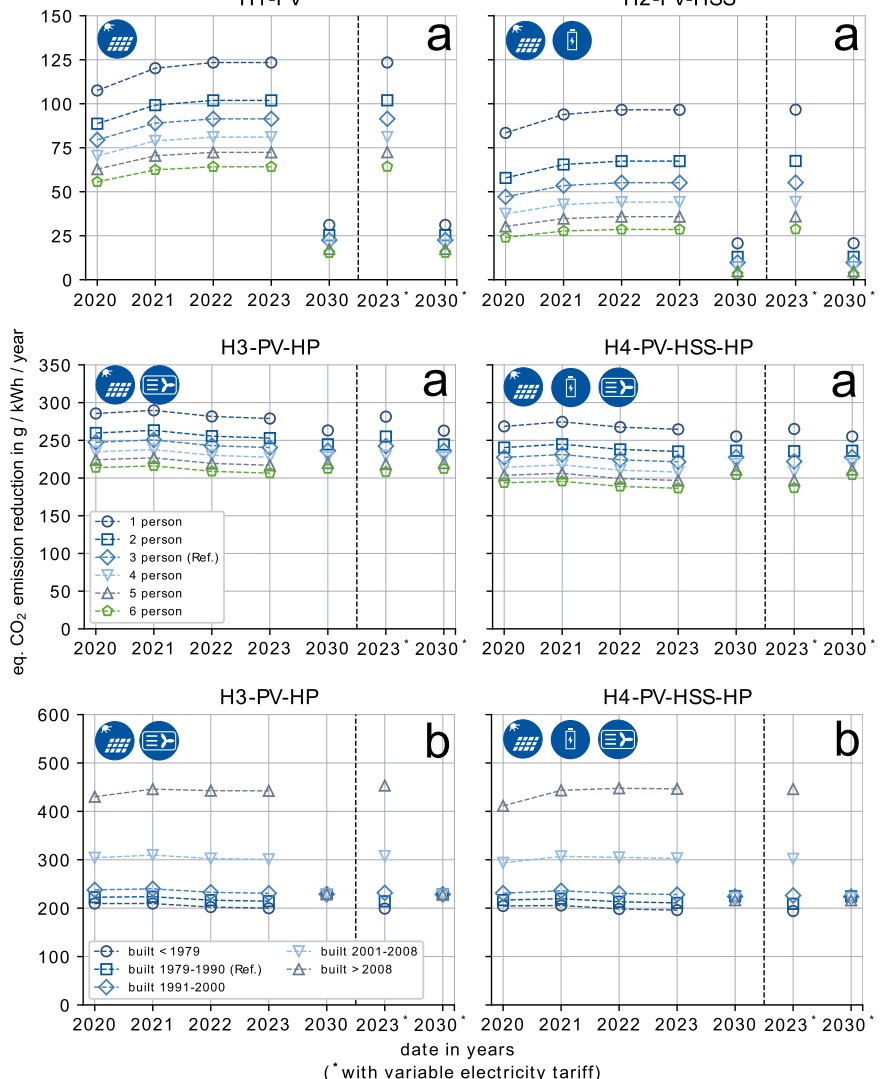

**Fig. 4 | Prosumer emission reductions.** Equivalent $CO_2$ reductions per kWh of consumed energy that can be achieved for different SFH topologies, compared to a standard fossil household utilizing NG. **a** Variation of number of residents, for a building constructed between 1979 and 1990. **b** Variation of construction year of a building, for a three-person household.

available rooftop area in Germany[30]. The cost savings results for the typical three-person household in Fig. 5a highlight that increasing the size of the solar PV system is usually beneficial. Especially for topologies with air-source HP (H2-PV-HP) the savings increase by 116–208 € (2020–2023). The topology with all renewable technologies combined (H4-PV-HSS-HP) benefits the most of a larger PV system as the additional generation can be flexibly consumed by utilizing the HSS or the hot water storage (HWS). For the topologies without air-source HP, however, the advantage is only small in 2020, and toward 2023, the larger PV system can even become financially less attractive. This is mainly because of the lower feed-in tariffs for the larger PV system in 2023. Furthermore, small additional savings that are generated by the increase in autarky of the household with the larger system are neutralized by the additional investment costs.

Regarding the equivalent $CO_2$ emissions of a SFH, a larger solar PV system does provide a substantial benefit compared to the smaller system (Fig. 5b). The results with or without a HSS are very similar. For the PV-only topology (H1-PV), the equivalent $CO_2$ emission reduction potential of a 13.7 kWp solar PV system is 76–87 g/kWh of consumed energy higher than for a 8.7 kWp system (2020–2023). This is simply caused by higher PV generation substituting high emissions from the

public electricity grid. The advantage of the topology with a HP system is similar. The reduction potential increases by 75–82 g/kWh (2020–2023). For 2030, the emission savings advantage of the larger system remains despite the ongoing decarbonization of the public electricity grid. Overall, the results support the statement that for most SFH configurations, for PV systems up to 13.7 kWp, a larger system provides benefits both financially and environmentally. However, for the PV only SFH a larger PV system can be less financially attractive in a few scenarios where self-consumption increase is minor compared to the higher investment costs.

**Influence of owning an EV on the results.** SFH residents who own an EV at the same time have a substantially higher electrical consumption with a certain charging characteristic. Therefore, we analyze the cost savings potential for SFH topologies that already own an electric vehicle when installing renewable components (H1–H4). This does not include any investment costs for an EV and additional electricity consumption for EV charging is assumed to be inflexible. The results in Fig. 5c summarize the savings potential for different SFH topologies with and without EV for different years. One key finding is that for EV owners, the benefit of investing in renewable components is always higher than for

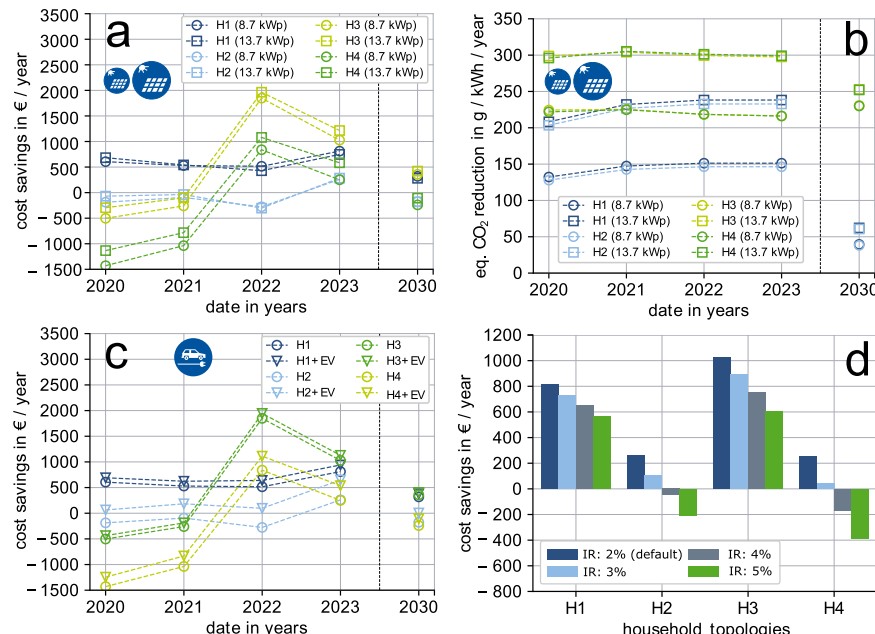

**Fig. 5 | Sensitivity analysis: influence of relevant modeling parameters on savings results, for a typical three-person household (constructed: 1979–1990) and different SFH topologies (H1–H4). a** Variation of PV size (costs results). **b** Variation of PV size (emission results). **c** Variation of demand—adding fixed charging schedule of an EV. **d** Variation of interest rate (investment costs) for 2023.

households without EV. For topologies with HSS, this benefit is particularly pronounced with 184–372 € increase in savings (H2 and H4). The observed effects mainly come from the higher utilization of solar PV generation as of increased consumption to hours without substantial solar radiation (evening or night). Therefore, especially with a HSS, the self-consumption can be further increased. Overall, with the projection that the number of SFH with EV will substantially increase[15], renewable investments also will become increasingly more attractive.

**Influence of component costs on the results.** In the aftermath of the lockdowns during the COVID-19 pandemic and with the emergence of the energy crisis there is a higher uncertainty in component costs. Therefore, to account for this uncertainty, we analyze the effects of an increase of interest rate between 2 and 5%. This leads to a 10–30% increase in investment costs for renewable technologies in 2023 (Fig. 5d). The results highlight that depending on the topology, the impact of the increase in interest rate on savings is differently pronounced. Investing in solar PV (H1-PV) remains a no-regret option as the increase in costs is distributed over a comparatively long lifetime. With an increase in interest rate to 5% (31% component costs increase) the annual cost savings decrease only by 30%. On the contrary, for the household topology with solar PV and a HSS (H2-PV-HSS) the higher investment costs lead to higher reductions of the savings. Even with only a 1% increase in interest rate (10% component costs increase) the savings drops by 59%. In general, topologies with HSS (H2 and H4) have higher investment costs with, at the same time, shorter component lifetimes and thus are more strongly affected by the increase in interest rate. For the fully renewable household topology (H4-PV-HSS-HP), an increase in the interest rate of 5% even leads to annual losses of 389 € in comparison with the standard fossil SFH (HO-GAS). For household topologies with an air-source HP, the savings results are less affected by the component costs increase and the investment remains highly attractive in 2023. One factor for this is that savings are highly dependent on the NG price and that the dimensioning of HP and electric top-up coil (TC) is dynamic. Therefore, the electric TC can take over some of the peak demand when investment costs for the HP increase.

## Results summary

In our study we compare different renewable energy systems of German residential Prosumers and quantify the impact of the energy crisis and regulations on their energy costs and $CO_2$ emissions. This is achieved by modeling different Prosumer topologies with the Framework for Optimizing Sector-Coupled Urban Energy Systems (FOCUS) which features a dynamic mixed-integer linear programming (MILP) Prosumer optimization model. Scenario variations include the comparison of different historical years (2020–2022), a current year (2023), and the year 2030 as a future outlook. Furthermore, relevant regulations are considered, including price breaks for electricity and NG, as well as VAT exemptions for renewable investments in 2023. Moreover, a wide range of typical SFH characteristics is covered within each scenario. For this purpose, five categories of SH demand are defined according to the construction year of the building, and the electricity and HW consumption is varied according to the number of residents, which ranges from one to six. The standard SFH consists of three residents in a building that was constructed between 1979 and 1990. The renewable topologies are compared with the standard fossil SFH utilizing NG for heating and cost savings as well as emission reductions are calculated accordingly.

Savings results prove that PV installations are economically beneficial in all of the studied years despite decreasing feed-in tariffs. The slightly decreasing savings of PV systems from 2020 to 2021 reverse in 2023 due to higher electricity prices caused by the energy crisis and the VAT exemption for PV investments. Combining a HSS with PV is economically less attractive than a PV-only household. This gap decreases during the energy crisis as the HSS reduces expensive grid supply. In 2020 and 2021, HP installations have a financial disadvantage over GB systems for all households built in 2008 or earlier. During the energy crisis, this changes when HP households achieve substantial savings of 1848 € (reference SFH in 2022) due to the sharp increase in the price of NG. Furthermore, the savings results indicate that a combination of solar PV, HSS, and HP is economically less attractive than the HP-only topology. For all technology combinations, results show that the savings of installing renewable technologies increases with the number of residents. On the other side, the year of construction has a major

influence on savings. During the energy crisis especially older and less efficient buildings can profit from installing an air-source HP.

The impact of variable electricity tariffs on the savings results proves to be relevant. For SFH with PV only and with additional HSS, cost savings decrease in 2023 as variable tariffs already reduce costs for the standard fossil SFH. For SFHs with HP, flexible pricing allows to consume electricity from the grid at lower prices, which increases cost savings in 2023 between 335 and 401 € for the reference SFH. However, for emission savings the impact of variable pricing is negligible. The same holds for the future 2030 scenario as lower electricity prices reduce possible savings by shifting the energy consumption of the HP. Another influence on the results can be observed with price breaks implemented on electricity and NG by the German government for 2023. They result in substantially lower cost savings of 40% for the SFH with HP as they effectively subsidize the standard fossil SFH. Despite the price breaks, all standard three-person SFH topologies still show substantial potential savings by installing renewable technologies.

The analysis of the reduction potential of $CO_2$ emissions indicates that all combinations of renewable technology lead to lower $CO_2$ emissions than the standard fossil household. The component with by far the largest reduction potential is the air-source HP with up to 251 g/kWh of energy (three-person standard SFH). The sole installation of a PV system already reduces the equivalent emissions by 79–91 g/kWh. Furthermore, with increasing number of residents, less reductions of equivalent emissions can be achieved as the consumption increases and PV generation is limited to the roof size. The highest savings can be achieved for energy-efficient buildings constructed after 2008 with reductions of up to 446 g/kWh per year.

In addition to the main results, we analyze the influence of relevant parameters on cost savings and $CO_2$ emission reductions. Increasing the size of the solar PV system (up to 13.7 kWp) usually results in higher cost savings (or equal costs) and lowers the annual equivalent $CO_2$ emissions per consumed kWh of energy. Furthermore, owning an EV increases self-consumption and results in higher savings for all topologies, which especially benefits topologies with installed HSS. Moreover, increasing the costs of renewable technologies by up to 30% in 2023 reduces the savings; however, different topologies are more or less affected. While the PV-only SFH topology is robust against increasing investment costs, the savings for topologies with HSS substantially decline.

Overall, the results indicate that the picture will substantially change by 2030. The cost savings and emission reductions with PV systems are projected to decrease with progressing decarbonization of the public electricity grid. Therefore, a reevaluation of remuneration mechanisms should be carefully considered in the coming years when aiming for steady growth rates of residential solar PV.

## Methods

There is a variety of possible configurations for the energy system in a SFH. The "Modeling and analysis procedure" section gives an overview of the considered household topologies and the general modeling procedure in this study. In addition, it provides the methodology for the comparison of costs and emissions of different household topologies for the analysis ("Savings potential with renewable investments in SFH" section and following). Moreover, in the "Optimization approach" section, the detailed modeling and optimization procedure is explained. Finally, the "Scenario set-up" section summarizes the methodology with a detailed overview of the ecological and economical parametrization for energy supply system and household components.

### Modeling and analysis procedure

The scenario set-up of this study considers five different topologies for SFH. We consider a standard fossil consumer household without any renewable installations and a GB for heating supply as standard fossil set-up (H0-GAS). By adding different renewable components to the base set-up we derive four household variations (H1–H4) (Fig. 6). All of these variations have in common that they include the installation of a solar PV system. This converts the SFH from a consumer to a Prosumer which has the ability to generate its own electricity. In addition to the solar PV-only Prosumer (H1-PV), the installation of a HSS (H2-PV-HSS) or switching the heating supply system from a GB to an air-source HP (H3-PV-HP) is considered. Finally, H4-PV-HSS-HP combines all renewable components. For all HSS, the operation is limited to self-consumption increase and the HSS cannot charge from or feed back into the grid. For all household configurations including the fossil topology (H0-GAS to H4-PV-HSS-HP) we optionally consider residents to own an EV. However, a comparison between electric vehicles and internal combustion engine (ICE) vehicles is not part of this study.

The scope of the analysis in this study is to make yearly comparisons between different topologies of SFH. The years under

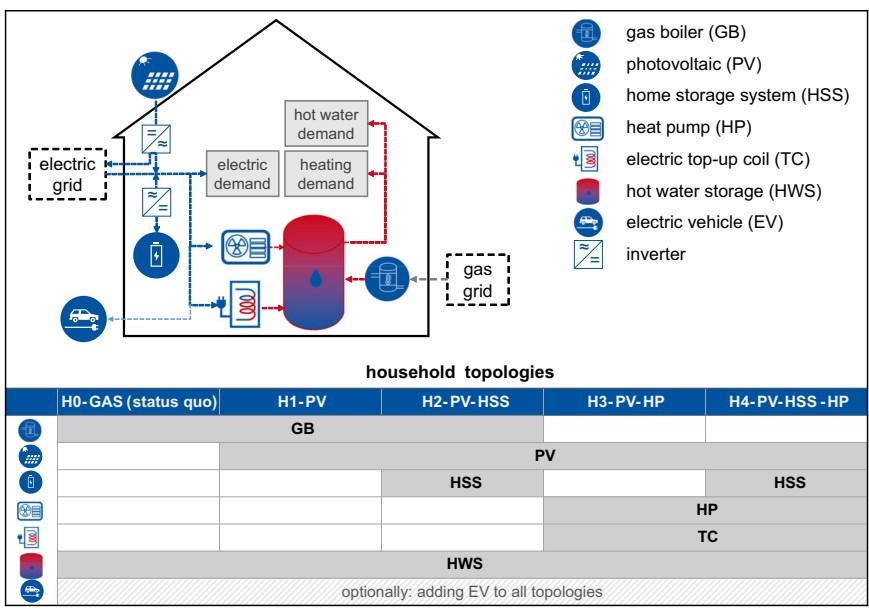

**Fig. 6 | Household topologies.** Definition of energy systems of selected SFH considered for the analysis in this study.

consideration include 2020 to 2023 to cover the conditions before and during the energy crisis, as well as 2030 to estimate future developments. The yearly energy flows of the SFHs are obtained by using FOCUS[31] which is a mathematical energy system modeling framework ("Optimization approach" section). From the optimized energy flows the key performance indicators (KPIs) for yearly costs ($S_{cost,a}$) and equivalent $CO_2$ emissions per consumed kWh of energy ($R_{emi,a}$) can be obtained. For the assessment of the savings and sustainability of renewable investments, we compare the different Prosumer topologies with the fossil consumer status quo set-up (H0-GAS). By subtracting the total costs of the renewable Prosumer topologies ($\pi_{Hj}$) from the costs of the fossil status quo SFH ($\pi_{H0-GAS}$), cost savings ($S_{cost,a}$) in EUR are calculated (Eq. (1)). Similarly, the reduction ($R$) of the emission factor $\epsilon$ in $gCO_2$ equivalents per consumed kWh of energy (electricity and NG combined) is obtained (2). Other KPIs that are used for the analysis include the self-consumption index (SCI) and the self-sufficiency index (SSI). While the SCI (Eq. (3)) represents the share of PV-generated electricity that is effectively consumed by the SFH ($E_{PV2SFH_{elec},Hj,a}$), the SSI (Eq. (4)) is defined as the share of combined total demand for heat and electricity ($E_{demand_{elec+heat},Hj,a}$) that is provided by self-generation through the PV system.

$$S_{cost,a} = (\pi_{H0-GAS,a} - \pi_{Hj,a})\ EUR \quad j \in \{1,2,3,4\} \quad a \in \{2020,...,2030\}$$
(1)

$$R_{emi,a} = (\epsilon_{H0-GAS,a} - \epsilon_{Hj,a})\ \frac{gCO2}{kWh} \quad j \in \{1,2,3,4\} \quad a \in \{2020,...,2030\}$$
(2)

$$SCI = \frac{E_{PV2demand_{elec},Hj,a}}{E_{PV,Hj,a}} \quad j \in \{1,2,3,4\} \quad a \in \{2020,...,2030\}$$
(3)

$$SSI = 1 - \frac{E_{grid_{elec},Hj,a} + E_{grid_{NG},Hj,a}}{E_{demand_{elec+heat},Hj,a}} \quad j \in \{1,2,3,4\} \quad a \in \{2020,...,2030\}$$
(4)

## Optimization approach

For the modeling and simulation of the SFH we use the FOCUS Framework[31] that has been developed at RWTH Aachen University. It features a Prosumer optimization model that is based on MILP. Supplementary Fig. 1 illustrates the information flow within the Prosumer model. In a first step, the user initializes the Prosumer class with required inputs, including Prosumer topology, components, and optimization objective. The Prosumer class then creates objects of the component class for all components included in the Prosumer topology (Fig. 6). While some components including solar PV systems are modeled with typical fixed sizes, most of the heating components are optimized in size. This includes the GB, the air-source HP, and the electric TC. This is required as the heating demand can be very different depending on the construction year of a building. Therefore, savings of components with comparably high investment costs, like an air-source HP, highly depend on the optimal size. In addition to the component class, the Prosumer class creates an instance of the energy management system (EMS) class in which the objective function is set up according to the desired optimization objective. All class objects write into the optimization model, which consists of the optimization variables, constraints, and the objective function. The model is then solved by the GUROBI solver[32]. Finally, energy flow, costs, and emission results are extracted from the optimized model.

The strategy for optimizing the SFH is to minimize the yearly overall system costs $\pi_{Hi,a}$ (Eq. (5)) for the respective year under consideration. The overall system costs for one year consist of capital

expenditures (CAPEX) and operational expenditures (OPEX). The CAPEX, in turn, include investment costs ($\pi_{cap,c}$) and fixed operational expenditures for maintenance ($\pi_{op,fix,c}$) of all components. For OPEX, we consider variable operating costs for purchasing energy ($\pi_{op,var,a}$) for one respective year. The total investment costs are annualized over a time horizon ($T$) of 20 years with interest rate $i$ whereas the variable operational costs are always calculated solely for the one respective year of consideration. The fixed operational costs for maintenance are calculated as fixed percentage ($f$) of the basic investment costs ($\pi_{cap,inv,npv,c}$). For total investment costs ($\pi_{cap,c}$), re-investments ($\pi_{cap,reinv,npv,c}$) for components that have a shorter lifetime than the time horizon of 20 years are considered. In addition, for components that have a longer lifetime than this horizon, residual values ($\pi_{cap,res,val,npv,c}$) are calculated for compensation. All costs components are calculated according to VDI2067[33], thus net present values (NPV) are summed up and multiplied by the annuity factor (Eq. (6)). For yearly OPEX that is added up in ($\pi_{op,var,a}$), energy purchasing costs for electricity ($\pi_{op,elec,a}$) and NG ($\pi_{op,gas,a}$) are taken into account. In addition, solar PV grid injection is reimbursed with fixed feed-in tariffs ($\pi_{op,PV,inj,a}$).

In contrast to system costs, emissions are not minimized. The equivalent $CO_2$ emissions are obtained from the optimized component sizes and energy flows. This includes $CO_2$ emissions from burning fossil NG in a GB ($\lambda_{GB,a}$), equivalent $CO_2$ emissions for provision of electricity ($\lambda_{grid,elec,out,a}$) and NG ($\lambda_{grid,gas,out,a}$) from the public grids, and equivalent $CO_2$ emissions for the manufacturing of components ($\lambda_{comp}$) including solar PV systems, HSS, and HP. In addition, emissions can be saved by feeding solar PV generation into the grid ($\lambda_{grid,elec,in,PV,a}$). We calculate a household emission factor ($\epsilon_{Hi}$) by dividing the total emissions by the total energy consumption of the household, which consists of annual heat ($\mathcal{E}_{heat,a}$) and electricity ($\mathcal{E}_{elec,a}$) consumption.

$$\begin{aligned}\pi_{Hi,a} &= \min(\pi_{CAPEX} + \pi_{OPEX})\ EUR \\ &= \min(\pi_{cap,c} + \pi_{op,fix,c} + \pi_{OPEX})\ EUR \\ &= \min(\pi_{cap,c} + \pi_{op,fix,c} + \pi_{op,var,a})\ EUR \\ &\forall c \in \{Components\} \quad \forall a \in \{Years\}\end{aligned}$$
(5)

$$\pi_{cap,c} = \frac{(1+i)^T * i}{(1+i)^T - 1} * t\left(\pi_{cap,inv,npv,c} + \pi_{cap,reinv,npv,c} - \pi_{cap,res,val,npv,c}\right)\ EUR$$
(6)

$$\pi_{op,fix,c} = f * \pi_{cap,inv,npv,c}\ EUR$$
(7)

$$\pi_{op,var,a} = (\pi_{op,elec,a} + \pi_{op,gas,a} - \pi_{op,PV,inj,a})\ EUR$$
(8)

$$\epsilon_{Hi,a} = \frac{(\lambda_{GB,a} + \lambda_{grid,elec,out,a} - \lambda_{grid,elec,in,PV,a} + \lambda_{grid,gas,out,a} + \lambda_{comp})\ gCO2}{(\mathcal{E}_{heat,a} + \mathcal{E}_{elec,a})\ kWh}$$
(9)

The standard configuration of the used Prosumer optimization model solves the entire MILP problem for one year and thus uses perfect foresight for all time series including renewable generation. In a first step, this configuration is used to obtain the optimal component sizes. However, in order to obtain results closer to the real operation a rolling horizon approach based on ref. 34 is used ("Rolling horizon procedure" section in Supplementary information). The approach reduces the optimization problem to smaller sub-problems (rolling horizon intervals) of 48 h. Starting with the first time step of the optimization, the first 48 h rolling horizon interval is initialized. For the first 24 h, the actual values of demand, solar irradiation, and temperature are used. For the last 24 h, prediction values are considered.

**Table 1 | Definition of scenarios for selected years from 2020 to 2030**

| | Unit | 2020 | 2021 | 2022 | 2023 | Future: 2030 | Reference |
|---|---|---|---|---|---|---|---|
| Electricity price | €ct/kWh | 31.81 | 32.16 | 38.57 | 46.91 | 23.12[d] | 12,38,39 |
| NG price | €ct/kWh | 5.97 | 7.06 | 16.48 | 16.11 | 6.47[d] | 12 |
| Ratio electricity/NG price | | 5.33 | 4.56 | 2.34 | 2.91 | 3.57 | 12 |
| Feed-in limit for solar PV | % of $P_N$ | 70 | 70 | 70 | 100 | 100 | 39 |
| Feed-in tariff for solar PV (8.7 kWp) | €ct/kWh | 9.87 | 8.16 | 6.83 | 8.2 | 7.22[d] | 36,52 |
| Feed-in tariff for solar PV (13.7 kWp) | €ct/kWh | 9.79 | 8.10 | 6.78 | 7.10 | 6.96[d] | 36,52 |
| Electricity price break (80% consumption) | €ct/kWh | 0 | 0 | 0 | 40 | 0 | 25 |
| Gas price break (80% consumption) | €ct/kWh | 0 | 0 | 0 | 12 | 0 | 25 |
| VAT PV and HSS | % | 19 | 19 | 19 | 0 | 19 | 36, 39 |
| VAT other components | % | 19 | 19 | 19 | 19 | 19 | 36, 39 |
| Avg. emission factor electricity grid | gCO₂eq./kWh$_{el}$ | 438 | 485 | 497 | 497[c] | 141 | 20,37,53 |
| Avg. emission factor NG provision | gCO₂eq./kWh$_{gas}$ | 28[d] | 28[d] | 19[d] | 16[d] | 55[d] | 27–29 |
| Emission factor burning NG | gCO₂/kWh$_{gas}$ | 202 | | | | 202 | 54 |
| Investment horizon | a | 20 | | | | 20 | |
| Interest rate | % | 3 | | | | 3[a] | 55 |
| Inflation rate[b] | % | 2 | | | | 2[a] | 56 |

[a]Assumption: same as 2023.
[b]Assumption: average of last 20 years.
[c]Assumption: same as 2022.
[d]Own calculation.

The predictions differ between the time series data (Supplementary Fig. 1) and consist of simple assumptions. For solar irradiation, for example, it is assumed that the values on the next day correspond to those on the current day (same hour last day) as the weather usually does not change rapidly. When the first 48 h rolling horizon interval is initialized, it is optimized with the FOCUS Framework and the first 24 h are taken as final results. The interval is then shifted by 24 h, and the process is iteratively continued over the course of one year. Between each shift, the storage levels of HSS and HWS are transferred. Before executing the rolling horizon layer the sizes of the heating components are pre-sized in a sizing step to match the peak loads. During this sizing step, the whole optimization problem of one year is solved once. This is necessary to avoid infeasibilities during the rolling horizon procedure.

## Scenario set-up

The analysis in this study distinguishes between five scenarios, each representing different years (Table 1). This includes a 2020 scenario as a benchmark before the energy crisis. However, the main focus is on the scenarios from 2021 to 2023 to capture the influence of the rising energy prices on SFHs. In addition, a fifth scenario for 2030 aims at predicting future developments. For all scenarios, we initially assume fixed tariffs for electricity and NG. However, since there is a growing number of flexible tariffs for electricity in Germany and extensive smart-meter roll-out is anticipated until 2030[35], for the 2023 and 2030 scenarios we also apply a flexible electricity tariff. The flexible tariffs are based on the 2023 stock exchange electricity prices (Day Ahead Auction)[2] and are normalized to the average prices of the fixed tariffs. Apart from energy prices, regulatory conditions are continuously changing and relief packages have been implemented by the government in 2023 to reduce the impacts of the energy crisis[25,36]. For 2023, we consider the following changes in regulations for the modeling:

- Adjustments of the feed-in tariffs for solar PV (see Table 1)
- Abolition of the 70% feed-in limit for the nominal power of residential solar PV generation
- Waiver by the legislator of the VAT for purchasing solar PV or a HSS

- Implementation of price breaks for electricity and NG with a fixed price for 80% of a household's consumption (with the previous year as a reference)

Additional investment grants for components, e.g., for HP systems are out of the scope of this study and are, therefore, not considered as they heavily depend on the boundary conditions of the household. Other factors that change between the yearly scenarios are emission-related. The equivalent emission from the electricity grid including upstream chains have seen an increase during the energy crisis[37] as coal-based generation increased. For NG provision, emission factors are obtained by weighting the supplier countries based emission factors for NG provision with the share of imports to Germany[27–29]. The factor has declined during the energy crisis as emission-rich NG imports from Russia have been reduced to zero until mid-2022. They were mainly replaced by imports from other European suppliers[29].

For the 2030 scenario, several assumptions are necessary. For electricity price, we consider reduced generation costs with high penetration of renewable energy sources[38], the elimination of the EEG levy[39], and constant taxes or other levies. The 2030 feed-in tariffs for solar PV are based on the updated regulation (Erneuerbares Energien Gesetz (EEG) 2023), which includes that tariffs will stay fixed until the beginning of 2024 and then decline by 2% per year[39]. Other assumptions for 2030 include that the relief packages of 2023, including VAT exemptions and price breaks, no longer exist. Furthermore, the emission factor of the electricity grid will substantially decrease due to higher penetration of renewable generation[20]. On the contrary, we assume that the emission factor for NG provision increases as it will mainly rely on imports of emission-rich liquified natural gas (LNG)[27].

In addition to a broad scenario scope, the aim is to cover a wide range of typical SFH characteristics in Germany. One parameter that mainly influences the characteristic of a SFH is the consumption behavior. For electricity and HW supply, the consumption is mainly influenced by the number of residents[40]. The higher the number of residents the higher is also the consumption (Table 2). The scope of this study is limited to a SFH with one to six residents (E1–E6).

**Table 2 | Classification of yearly electrical and HW consumption of German SFH, derived from ref. 40**

| | | Electricity and HW demand of typical German SFH | | | | | | |
|---|---|---|---|---|---|---|---|---|
| | | E1 | E2 | E3 | E4 | E5 | E6 | Reference |
| Type | | Single | Couple | Family | | | | |
| Number of residents | | 1 Person | 2 Person | 3 Person | 4 Person | 5 Person | 6 Person | 40,57 |
| HW consumption (NG) | kWh/a | 500 | 1000 | 1500 | 2000 | 2500 | 3000 | 40,44,58 |
| Electricity consumption | kWh/a | 2350 | 4040 | 4950 | 6000 | 7000 | 8100 | 40 |
| Electricity consumption EV charger | kWh/a | 3210 | | | | | | 15,40–42 |
| Rooftop PV-potential | kWp | 8.7/13.7 | | | | | | 30 |

**Table 3 | Classification of yearly SH consumption of German SFH, partly derived from refs. 43,44**

| | | Space heating demand of typical German SFH | | | | | |
|---|---|---|---|---|---|---|---|
| | | TH1 | TH2 | TH3 | TH4 | TH5 | Reference |
| Type | | Unrenovated housing stock | | | Well-insulated | Low-energy | |
| Construction year | | <1979 | 1979–1990 | 1991–2000 | 2001–2008 | >2008 | 40,57 |
| Efficiency class | Scale | G | F | E | C/B | A+ | 59 |
| Share of German SFH | % | 16 | 15 | 13 | 12 | 5 | 46 |
| Living area (avg) | m² | 108 | 122 | 131 | 138 | 149 | 43,44,57 |
| Heating consumption (avg) | kWh/m²/a | 244 | 185 | 146 | 75 | 25 | 40,57 |
| Heating demand (avg) | kWh/a | 26,352 | 22,570 | 19,126 | 10,350 | 3725 | Calculation |
| Total gas demand (avg) | kWh/a | 28,007 | 23,988 | 20,327 | 11,000 | 3959 | Calculation |

Furthermore, it is assumed that HW consumption is provided by a GB using NG for the fossil standard household topology. For residents that own an EV, we use an internal simulation tool at ISEA RWTH Aachen based on ref. 41 to create a charging profile for a typical 11 kW AC home charger. We consider an EV with a battery capacity of 35 kWh (German average in 2022) and an average consumption of 20 kWh/100 km[15]. From the simulation, we observe a yearly driving distance of 14,610 km per year which fits the typical driving characteristic of a German car owner[42].

For SH demand, the construction year of the building is the most influential factor. Therefore, we define five different SH demand categories (H1–H5) with a consumption ranging from over 26,000 kWh to below 4000 kWh per year (Table 3), partly derived from refs. 43,44. NG demand is then calculated with a GB efficiency of 96% from the SH demand[45]. The combination of different consumption patterns for electricity, SH, and HW allows for a comprehensive analysis of the potential of saving measures for a wide range of SFH in Germany which make up about 68% of the building stock[44]. The average heating demand in SFH in Germany is 175 kWh/m2/a, and the average household size is assumed to be 3 residents[44,46]. Therefore, the standard household for this study is parameterized with heating demand TH2 and electrical and HW demand of E3.

As input data, the optimization model requires the energy consumption for electricity, SH, and HW for each time step to be able to optimize the energy flows. Therefore, yearly demands have to be converted into time series data. The temporal resolution is chosen to be 15 minutes which is a common measurement resolution in the European energy sector. For all consumption types, a typical demand profile is defined which is then normalized to the required yearly consumption (Supplementary Table 2). We assume that the energy consumption of all households remains the same for all considered years to be able to isolate the effects of renewable investments. Therefore, we do not account for the energy savings that have been achieved during the energy crisis[7,12]. In addition to demand data, for irradiation data, the average measurements for Germany in 2019 are taken from the website *renewables.ninja*[47], which are based on

MERRA-2[48,49]. For temperature data, the test reference years from the German weather service (DWD) are used[50] as the SH and HW demand data is based upon them. As the resolution of the used input data varies it is re-sampled to a 15-min resolution for all time series. The quality of the results from the optimization procedure strongly depends on the detailed parameterization of component costs, sizes, and efficiencies. Supplementary Table 3 summarizes the technical and economical data that is used for the modeling of the required components. A detailed description of the used component data can be found in the "Component modeling" section in Supplementary information.

## Data availability
The optimized Prosumer energy flow data generated in this study have been deposited in the RWTH Publications database[51] and can be found under the following https://doi.org/10.18154/RWTH-2024-07163. Further, the figures data are provided in the Supplementary Information/ Source Data file. For more information on the simulation data, please contact batteries@isea.rwth-aachen.de.

## Code availability
The source code of the underlying FOCUS model used for the analysis within this work is available in the following code repository: https:// git-ce.rwth-aachen.de/focus.

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

## Acknowledgements

The research for this paper was performed within the InEEd-DC project supported by the German Federal Ministry of Education and Research under grant number 03SF0597 and has been received by J.O., M.C., N.N., J.G., J.F., S.Z., C.B., and D.S.

## Author contributions

Jonas van Ouwerkerk: Conceptualization, Methodology, Software, Validation, Formal analysis, Investigation, Data curation, Writing—Ori-ginal draft preparation, Writing—Review & Editing, Visualization, Pro-ject administration. Mauricio Celi Cortés: Methodology, Software, Validation, Data curation, Writing—Review & Editing. Najet Nsir: Con-ceptualization, Methodology, Software, Validation, Formal analysis, Investigation, Data curation, Writing—Original draft preparation, Writ-ing—Review & Editing, Visualization. Jingyu Gong: Software, Valida-tion, Data curation, Writing—Review & Editing. Jan Figgener: Conceptualization, Writing—Review & Editing, Supervision, Project administration, Funding acquisition. Sebastian Zurmühlen: Con-ceptualization, Supervision, Project administration, Funding acquisi-tion. Christian Bußar: Conceptualization, Writing—Review & Editing, Supervision, Project administration, Funding acquisition. Dirk Uwe Sauer: Project administration, Funding acquisition.

## Funding

## Competing interests

The authors declare no competing interests.
