## [Peer Review File · Nature Communications]

Quantifying the benefits of renewable investments for German residential Prosumers in times of volatile and uncertain energy marketsReviewers' Comments:

Reviewer #1:

Remarks to the Author:

This study provides a cost and emissions analysis of German prosumers during the recent European energy crisis caused by the sudden removal of Russian natural gas. The results show that PV reduces costs across all years tested, whereas heat pumps and batteries vary by year and household occupancy. All technologies reduce emissions with the most effective being PV+HP.

The study is highly relevant and useful for the current status of Europe's energy system – up-to-date descriptions on the techno-economic status of prosumer energy systems are important at all times, but in particular under the current uncertainty and volatility. Generally speaking, the approach is well treated, but there are a few methodological points that should be addressed prior to publication.

1. In Ch.2 it is not clear what the cost savings per year represents. The way the results are presented in line graphs and the wording at times suggests the savings are operational savings in specific years absent any capital costs. However the description in Ch.6.2 states that it also includes annualized capital costs and fixed operational costs, suggesting these values are more of an annualized total life cycle cost based on 20 year investment lifetimes. If this is the case, are the results only using prices from the stated year applied for the full 20 year investment? Or are investment prices taken from that year and the operational prices changing dynamically over the lifetime? Please clarify the KPI definition in Ch.2 and further clarify the method in Ch.6.

2. Both the cost and emissions savings are given in absolute values, however it would be valuable to also have relative values in percentage. This will help describe the impact of each treatment and provide more context to the results.

3. It would be valuable to have some standard prosumer KPIs like solar fraction and self-consumption provided in Ch.6 (perhaps Table 2 or 3). This aspect is discussed at points (for example Page 5, Line 134) as having an impact on the results, and a full set of results with these KPIs will help the reader understand the dynamics impacting the cost savings potential or recommended design of prosumer technologies.

4. Leading from point 3, please provide further motivation why the PV and battery capacities are not included in the optimization as the HP, TC, and GB are. I don't mean to say the approach is incorrect, just that it is an open question to the reader why some component capacities are optimized and others are not that should be explained. An alternative approach would be to keep the solar fraction fixed, leading to a different PV capacity for each household. It would also be valuable to include the HP and TC capacities for each building type in a results table in Ch.6 both for documentation and in case there is a design guideline that can be offered by the results.

5. Throughout the paper, the terms "cost savings" and "profitability" are used interchangeably. I would argue these are different things and describing profitability require KPIs not used in the

present study (e.g. IRR or ROI). When used to generally refer to a system configuration option as being more/less profitable, this will probably map correctly to the system with positive/negative cost savings. However, there are few points (e.g. Page 5, Line 143; Page 8, Line 264; Page 9, Line 296, Page 9, Line 300) where specific cost savings values are labeled as increases/decreases of profitability, and I would argue this is an incorrect use of the term. Without the use of an investment metric like IRR, ROI or PI, my suggestion is to limit the discussion of economic results in terms of cost savings and avoid the term “profitability” altogether. This would also fit better with the paper’s title.

Some minor, specific comments

1. Page 3, Lines 83-84: The phrase “temperature adjustments” sounds as though it could also refer to indoor temperature reductions to save energy/money. Suggest rephrasing to “...considering normal year temperature corrections.”
2. Page 10, Line 305: The term “heating rod” is used for the first time here. It is unclear what it refers to but in Ch.6 it becomes clear this is most likely referring to the electric top-up coil described in Figure 10. Suggest using a single term throughout the paper, preferable the top-up coil since it is already in the figure.
3. Page 11, Line 321: Typo, “proof” should be “prove”
4. Page 11, Line 337: Suggest rephrasing “The component with the by far largest...” to “The component with by far the largest...”

Reviewer #2:

Remarks to the Author:

It is a well-written paper which provides a sufficient background, clear methodology, and (mostly) sound and reasonable results and conclusions. The paper pinpoints the considerable effect of PV only as well as Air Source heat pump systems in saving cost for the households and reducing emission.

The work is original and has a scientific contribution. however, there are a few issues to be fixed before being able to publish it.

1. The paper repeatedly claims that Increasing the size of the solar PV system always results in higher cost savings. but this is not fully correct and might give a completely wrong conclusion to the readers. If PV only scenario is used, larger size of PV leads to much higher capital cost as well as lower self-consumption. That means the cost effectiveness of the investment will be at the mercy of feed-in-tariff. if the feed-in-tariff is not high enough, the cost effectiveness of large size of PV will be much lower and it is better not to oversize the PV system.

if you go for the scenario of HP+HSS, larger size of PV will require large size of battery which makes

the economic outcomes much poorer. in the case of HP + PV, there is still an optimum size of PV depending on the size of heat pump plus the consumption behavior.

so, to be short, I suggest that you rewrite the conclusion part of "increasing the size of PV" and correct the statements which might lead to serious misunderstanding of how to size PV system.

2. the electricity to gas price ratio is the elephant in the room in the paper. Why doesn't the research study the effect of electricity to gas price ratio on the results and conclusions?

3. The paper claims that adding EV always leads to economic benefit. Let's take an example of PV+EV scenario (without HSS). If the household has a variable electricity price tariff, it is usually economically more beneficial to charge EV batteries during the late nights and early hours of the day when the electricity price is much lower. If the user wants to take the benefit of higher self-consumption through charging EV batteries by PV electricity, they have to charge the EV battery in the middle of the day when the electricity price is much higher (even we assume that the car is at home at that time). That can easily lead to even higher cost for the user.

4. the economic results in the paper would be highly sensitive to interest rate. The sensitivity analysis should include interest rate.

Reviewer 1

1. In Ch.2 it is not clear what the cost savings per year represents. The way the results are presented in line graphs and the wording at times suggests the savings are operational savings in specific years absent any capital costs. However the description in Ch.6.2 states that it also includes annualized capital costs and fixed operational costs, suggesting these values are more of an annualized total life cycle cost based on 20 year investment lifetimes. If this is the case, are the results only using prices from the stated year applied for the full 20 year investment? Or are investment prices taken from that year and the operational prices changing dynamically over the lifetime? Please clarify the KPI definition in Ch.2 and further clarify the method in Ch.6.

→ The costs include investment as well as operational costs. Investment costs are annualized for the investment horizon of 20 years, including reinvestment and residual values. Fixed operational costs for maintenance are calculated based on a fixed percentage of the annualized investment costs and are added to CAPEX. For each of the years that we analyze in the study investment costs and fixed maintenance costs are calculated individually based on component prices. For variable operational costs, however, we only calculate over the duration of one year with the respective prices for electricity and natural gas. The overall costs that we compare in the end in terms of savings represent the energy costs that a household needs to pay during a respective year.

We rephrased the paragraph in Ch. 6.2 from line 429 and hope this is clearer now. Further, we updated equations 5-7 to make clear what cost components are considered and how we define OPEX and CAPEX. For further reference, we show the CAPEX and OPEX for all results of our base analysis in Appendix E.11 and E.12 at the end of the document. In Chapter 2, we included the description of how costs are calculated at the beginning (see line 119f).

2. Both the cost and emissions savings are given in absolute values, however it would be valuable to also have relative values in percentage. This will help describe the impact of each treatment and provide more context to the results.

→ We have included percentage values in the Appendix (E.13 and E.14) and refer to them in our analysis.

3. It would be valuable to have some standard prosumer KPIs like solar fraction and self-consumption provided in Ch.6 (perhaps Table 2 or 3). This aspect is discussed at points (for example Page 5, Line 134) as having an impact on the results, and a full set of results with these KPIs will help the reader understand the dynamics impacting the cost savings potential or recommended design of prosumer technologies.

→ We have included the KPIs values for the self-consumption index (SCI) and self-sufficiency index (SSI) in Appendix (E.9 and E.10) and reference them in our analysis in Section 2. Further, we have defined the KPIs in Chapter 6.1 as equations 3) and 4).

4. Leading from point 3, please provide further motivation why the PV and battery capacities are not included in the optimization as the HP, TC, and GB are. I don't mean to say the approach is incorrect, just that it is an open question to the reader why some component capacities are optimized and others are not that should be explained. An alternative approach would be to keep the solar fraction fixed, leading to a different PV capacity for each household. It would also be valuable to include the HP and TC capacities for each building type in a results table in Ch.6 both for documentation and in case there is a design guideline that can be offered by the results.

→ Generally, we tried to focus on the important variations of parameters and to reduce the variation wherever possible. Otherwise, the number of combinations would increase too much and the analysis would be very difficult. We chose to keep PV and HSS sizes rather fixed for the following reasons:

For PV, previous studies (Reference 33) have shown that the roof space of single-family homes is limited and therefore PV Systems have a very similar size between 8.7-13.7 kWp (which is also the input for our sensitivity analysis in Section 4). Moreover, as our study shows, PV is mostly economic and therefore it is usually beneficial to cover as much roof area as possible, which leads to quite similar system sizes of residential PV in Germany. Therefore, a stronger variation is in our opinion not needed to cover most of the single-family households in Germany with our analysis.

For home storage systems in Germany, the typical size is around 10kWh as the main motivation to buy a system is not purely economic but to help the energy transition and increase the autarky of the household (Reference 21). Thus, the investment in a home storage system is irrational from an investment perspective and most people buy a system close to 10 kWh anyway, which we wanted to cover in our analysis. Therefore, we decided to keep the size of the HSS constant.

In Appendix A (line 845f.) we added a few sentences for clarification.

5. Throughout the paper, the terms “cost savings” and “profitability” are used interchangeably. I would argue these are different things and describing profitability require KPIs not used in the present study (e.g. IRR or ROI). When used to generally refer to a system configuration option as being more/less profitable, this will probably map correctly to the system with positive/negative cost savings. However, there are few points (e.g. Page 5, Line 143; Page 8, Line 264; Page 9, Line 296, Page 9, Line 300) where specific cost savings values are labeled as increases/decreases of profitability, and I would argue this is an incorrect use of the term. Without the use of an investment metric like IRR, ROI or PI, my suggestion is to limit the discussion of economic results in terms of cost savings and avoid the term “profitability” altogether. This would also fit better with the paper’s title.

→ We fully agree. Our results do not support the wording. We changed all occurrences of profitability with savings and profitable with economically attractive.

Some minor, specific comments

1. Page 3, Lines 83-84: The phrase “temperature adjustments” sounds as though it could also refer to indoor temperature reductions to save energy/money. Suggest rephrasing to “...considering normal year temperature corrections.”

→ We corrected it by sticking to your suggestion.

2. Page 10, Line 305: The term “heating rod” is used for the first time here. It is unclear what it refers to but in Ch.6 it becomes clear this is most likely referring to the electric top-up coil described in Figure 10. Suggest using a single term throughout the paper, preferable the top-up coil since it is already in the figure.

→ We corrected it and are now only using the term top-up coil.

3. Page 11, Line 321: Typo, “proof” should be “prove”

→ We corrected this.

4. Page 11, Line 337: Suggest rephrasing “The component with the by far largest...” to “The component with by far the largest...”

→ We corrected it by sticking to your suggestion.

Reviewer 2

1. The paper repeatedly claims that Increasing the size of the solar PV system always results in higher cost savings. but this is not fully correct and might give a completely wrong conclusion to the readers. If PV only scenario is used, larger size of PV leads to much higher capital cost as well as lower self-consumption. That means the cost effectiveness of the investment will be at the mercy of feed-in-tariff. if the feed-in-tariff is not high enough, the cost effectiveness of large size of PV will be much lower and it is better not to oversize the PV system.

if you go for the scenario of HP+HSS, larger size of PV will require large size of battery which makes the economic outcomes much poorer. in the case of HP + PV, there is still an optimum size of PV depending on the size of heat pump plus the consumption behavior.

so, to be short, I suggest that you rewrite the conclusion part of "increasing the size of PV" and correct the statements which might lead to serious misunderstanding of how to size PV system.

→ We agree with you that this cannot be generally stated for all PV System sizes and rephrased our conclusion by mentioning that this does only apply in the certain range that we analysed. Increasing the PV size within the range from 8.7 to 13.7 kWp, as we did in this paper, is usually beneficial and leads to higher savings (see Chapter 4.1, line 313f / Chapter 5, line 383f./Abstract, line 21/Highlights, line 31).

2. The electricity to gas price ratio is the elephant in the room in the paper. Why doesn't the research study the effect of electricity to gas price ratio on the results and conclusions?

→ This is a very interesting idea and we see that this ratio is important. To highlight, that we are already considering it we added this ratio in Table 1. The values show, that this ratio is highly dependent on the respective year. It varies substantially between 5.3 and 2.3. Nevertheless, there is also always a coupling of electricity and natural gas prices as peak power plants in Germany (which are running on natural gas) in the end significantly influence the electricity price. In addition, there is a delaying effect as end customers have long-term contracts and thus see price developments at the stock exchange much later. Furthermore, not only the ratio but also the height of prices has an influence. The ratio for 2030 is not too different from that in 2023 but the savings results are substantially different for both years. In summary, the ratio is important but we conclude that it cannot explain the results alone. There are just many influential factors for each single year, like other regulatory constraints, that play a key role. Considering a broad variation of this ratio would lead to numerous combinations that would be very difficult to analyse in such a dense paper. This is why we focused on realistic scenarios and electricity to gas price ratios instead by considering typical past, present, and future years. I hope this clarifies our procedure. To add value to the analysis, we now included the ratio in the discussion part of the study (Chapter 2.3, line 178, line 187).

3. The paper claims that adding EV always leads to economic benefit. Let's take an example of PV+EV scenario (without HSS). If the household has a variable electricity price tariff, it is usually economically more beneficial to charge EV batteries during the late nights and early hours of the day when the electricity price is much lower. If the user wants to take the benefit of higher self-consumption through charging EV batteries by PV electricity, they have to charge the EV battery in the middle of the day when the electricity price is much higher (even we assume that the car is at home at that time). That can easily lead to even higher cost for the user.

→ We see that the conclusion regarding EV might be misleading. We do not consider any investment costs for the EV and only compare households that already have an EV before they install any renewable components. Therefore, we changed the phrasing to clarify this (Chapter 4.2, line 324).

The other topic you refer to is variable pricing and demand side management (flexible demand). We originally have not considered it as it is still rarely used in Germany. However, we agree on the importance and therefore included variable pricing for 2023 and 2030 in all scenarios of our base analysis (Chapter 2.5/ Chapter 3, line 288/Chapter 5, line 378/ Chapter 6.3, line 496f) to see the general trends. We find that the influence of variable pricing on the results, however, is only minor without considering demand side management (residents do not change their consumption behaviour). Only for the Heat Pump scenario, there are minor advantages.

Although it would certainly be interesting to consider demand side management, we decided that we do not look further into it at this point, as this would be out of scope of this work. The topic is highly complex and would be a paper on its own. Thus, for EV the demand remains static in our analysis. For this reason, and because the purpose of the sensitivity analysis with EV is simply to add additional fixed demand and see the effects on the base results, we did not further look into variable pricing in combination with EV. I hope still that the additional analysis of flexible pricing in our base analysis adds enough value to answer your questions in this regard.

4. the economic results in the paper would be highly sensitive to interest rate. The sensitivity analysis should include interest rate.

→ We fully agree and changed the analysis of component costs increase in the sensitivity analysis. Now this analysis considers interest rate variations from 2-5% (see Chapter 4.3 and Figure 5d).

General comments

- We found a small calculation error for the 2030 battery costs (line 853, 627 €/kWh). We corrected this and recalculated all 2030 scenarios with HSS.
- We clarified that the HSS is only used for self-consumption increase and cannot feed back into the grid (Chapter 6.1, line 422).

Reviewers' Comments:

Reviewer #1:

Remarks to the Author:

Thank you for your responses and edits to the paper, all of the recommendations have been satisfactorily addressed. It is a well executed study and in my opinion ready for publication.